# The Tetraspanin CD9 Facilitates SARS-CoV-2 Infection and Brings Together Different Host Proteins Involved in SARS-CoV-2 Attachment and Entry into Host Cells

**DOI:** 10.3390/v17081141

**Published:** 2025-08-20

**Authors:** Vanessa Rivero, María Laura Saiz, Daniel Torralba, Carlos López-Larrea, Beatriz Suarez-Alvarez, Marta L. DeDiego

**Affiliations:** 1Department of Molecular and Cell Biology, Centro Nacional de Biotecnología-Consejo Superior de Investigaciones Científicas (CNB-CSIC), 28049 Madrid, Spain; vrivero@cnb.csic.es; 2Translational Immunology, Health Research Institute of the Principality of Asturias (ISPA), 33011 Oviedo, Spain; mlsaizalvarez@gmail.com (M.L.S.); inmuno@hca.es (C.L.-L.); beatriz.suarez@ispasturias.es (B.S.-A.); 3Kidney Disease Research Network, RICORS2040, Instituto de Salud Carlos III (ISCIII), Madrid, Spain; 4PharmaMar S.A., Colmenar Viejo, 28770 Madrid, Spain; datorgra@gmail.com

**Keywords:** CD9, SARS-CoV-2, virus entry, virus-host interactions, proteases TMPRSS2 and furin, ACE2, NRP1, tetraspanins

## Abstract

CD9 protein belongs to a family of proteins called tetraspanins, so named for their four-transmembrane-spanning architectures. These proteins are located in domains in the plasmatic membrane, called tetraspanin-enriched microdomains (TEMs). Several proteases and cellular receptors for virus entry cluster into TEMs, suggesting that TEMs are preferred virus entry portals. Severe Acute Respiratory Syndrome coronavirus 2 (SARS-CoV-2) spike (S) protein mediates virus attachment and entry into cells by binding to human angiotensin-converting enzyme 2 (ACE-2). In addition, the secretory, type-I membrane-bound SARS-CoV-2 S protein is synthesized as a precursor (proS) that undergoes posttranslational cleavages by host cell proteases, such as furin and TMPRSS2. Moreover, it has been shown that neuropilin-1 (NRP1), which is known to bind furin-cleaved substrates, potentiates SARS-CoV-2 infectivity. Our results indicate that CD9 facilitates SARS-CoV-2 infection. In addition, we show how knocking out CD9 leads to a decrease in the expression of NRP1, a protein that improves SARS-CoV-2 infection. Furthermore, we show that CD9 colocalizes with ACE-2, NRP1, furin, and TMPRSS2 at the plasma membrane; that the absence of CD9 decreases the expression of these proteins on the plasma membrane CD9-enriched microdomains, and that CD9 interacts with ACE2. In conclusion, our data suggest that CD9 facilitates SARS-CoV-2 infection and that CD9 brings together different host proteins involved in SARS-CoV-2 attachment and entry into host cells, such as ACE2, NRP1, furin, and TMPRSS2. Importantly, the fact that a blocking antibody targeting CD9 can effectively reduce SARS-CoV-2 titers highlights not only the mechanistic role of CD9 in viral entry but also offers translational potential, suggesting that tetraspanin-targeting antibodies could be developed as therapeutic agents against SARS-CoV-2 and possibly other coronaviruses, with meaningful implications for clinical intervention.

## 1. Introduction

Enveloped viruses, such as Severe Acute Respiratory Syndrome coronavirus 2 (SARS-CoV-2), require the fusion of viral and host membranes to deliver the viral genetic material inside the cells, initiating viral infections. In the case of SARS-CoV-2, this process involves the binding of the spike (S) protein to its receptor, angiotensin-converting enzyme 2 (ACE2) [1,2,3,4,5,6], and the proteolytic activation of the S protein, which is divided into two domains, named S1 and S2. For this proteolytic activation, two proteolytic cleavage steps following ACE2 engagement are needed. The first of these is located at the S1–S2 boundary, and the second is localized to the S2′ site in the S2 subunit. In the case of SARS-CoV-2, the S1–S2 boundary contains a multibasic site (Arg-Arg-Ala-Arg) that is cleaved by furin in the virus-producer cell [3,4,7,8,9,10,11]. After the S1–S2 boundary is cleaved, the S2′ site must also be proteolytically cleaved to fully activate the fusion process either by proteases of the type II transmembrane serine protease (TTSP) family, mainly the TTSP member transmembrane protease serine 2 (TMPRSS2) on the cell surface [12,13,14], or by cathepsins in the endosomes [15]. Cell entry by SARS-CoV-2 is therefore dependent on the target-cell proteases, and TMPRSS2 and cathepsin L are the two major proteases involved in S protein activation. As TMPRSS2 is present at the cell surface, TMPRSS2-mediated S protein activation occurs at the plasma membrane, whereas cathepsin-mediated activation occurs in the endolysosome (reviewed in [16]). In addition to the proteolytic cleavages by proteases, neuropilin 1 (NRP1) was shown to enhance TMPRSS2-mediated entry of SARS-CoV-2 [17,18]. NRP1 was also shown to bind S1 through the multibasic furin-cleavage site and to promote S1 shedding and to expose the S2′ site to TMPRSS2 [19].

CD9 belongs to a family of proteins called tetraspanins (Tspans). Tspans are small proteins embedded in the membrane through four transmembrane domains, belonging to the large tetraspanin family [20]. Tspans contain four α-helical transmembrane (TM) regions (TM1–TM4), two extracellular (EC) domains (EC1 and EC2), and three cytoplasmic (CP) regions, comprising N-terminus, C-terminus, and a short loop that connects TM2 and TM3. These topological domains contribute differently to the structure of Tspans and to their ability to undergo intramolecular and intermolecular interactions [20], interacting with membrane-associated proteins, including cell receptors, transmembrane proteases, and other Tspans, to provide platforms critical for viral entry [21,22,23]. Tspan-enriched microdomains (TEMs) are associated with the entry sites of several viruses, including coronaviruses (reviewed in [22]), as TEMs contain the CoV receptors dipeptidyl peptidase 4 (DPP4), which is the receptor for Middle East Respiratory Syndrome coronavirus (MERS-CoV) [24], aminopeptidase N (APN), the receptor for HCoV-229E, ACE2, the receptor for SARS-CoV and SARS-CoV-2 [1,2,3,4,5,6], and APN and DPP4 [25]. For example, the Tspan CD9 positions DPP4, APN, and TMPRSS2 into TEMs, facilitating MERS-CoV and HCoV-229E entry [26]. The Tspans CD9, CD63, and CD81 facilitate the entry of the CoVs murine hepatitis virus (MHV), MERS-CoV, HCoV-229E, and SARS-CoV-2 [25], and Tspan8 facilitates SARS-CoV-2 infection rates independently of ACE2-Spike interaction [27]. Related to other viruses, the Tspans CD9, CD63, and CD81 facilitate the entry of influenza A virus (IAV) [25], tetraspanin CD151 mediates papillomavirus endocytosis [28], tetraspanin CD81 mediates hepatitis C virus (HCV) entry [29], and Tspans CD9 and CD81 modulate HIV-1-induced membrane fusion [30]. In addition, CD81 plays a role during influenza virus budding, as influenza virus infection recruits CD81 on the plasma membrane to the concentrated budding sites where other viral proteins are located [31].

In this work, we evaluated the role of the Tspan CD9 on SARS-CoV-2 infection. Using CD9 KO cells and anti-CD9 antibody, we found that CD9 positively affects SARS-CoV-2 infection. In addition, we show how knocking out CD9 leads to a decrease in the expression of NRP1, a protein that improves SARS-CoV-2 infection. Furthermore, we show that CD9 colocalizes with ACE-2, NRP1, furin, and TMPRSS2 at the plasma membrane, and that CD9 is present in the same plasmatic membrane domains as ACE-2, NRP1, furin, and TMPRSS2, leading the absence of CD9 to a decrease in the expression of these proteins on the CD9-enriched microdomains.

## 2. Materials and Methods

Cells and viruses. SW480 WT (ATCC CCL-228), A549 (ATCC CCL-185) cells, 293T cells (ATCC CRL-1573), and Vero E6 (ATCC CRL-1586) cells were obtained from the ATCC. All the cells were grown at 37 °C in air enriched with 5% CO_2_ using Dulbecco’s modified Eagle’s medium (DMEM, Gibco, Grand Island, NY, USA) supplemented with 10% fetal bovine serum (Gibco) and 50 mg/mL gentamicin (Gibco).

A549 and SW480 CD9 knock-out cells were generated using CRISPR-Cas9 guide RNAs designed with the CRISPOR tool (Version 5.2) [32]. Briefly, three candidate guide RNAs (gRNAs) targeting distinct exons (exons 1, 3, and 4) of the *CD9* gene were designed using the CRISPOR platform. After in silico evaluation of potential off-target effects and on-target scores, the guide targeting exon 1 was selected for further experiments due to its optimal balance between predicted efficiency and minimal off-target activity. The selected guide sequence for exon 1 was GCCCTCACCATGCCGGTCAA, adjacent to the protospacer adjacent motif (PAM) AGG. The PAM motif is critical as it is required for the Cas9 endonuclease to bind the target DNA and introduce a double-strand break three nucleotides upstream of the PAM. The CRISPR/Cas9 machinery, including the gRNA and Cas9, was co-transfected into the target cells using a plasmid that also expresses GFP as a reporter. This allowed for the selection of successfully transfected cells based on GFP expression via fluorescence-activated cell sorting (FACS). GFP-positive cells were isolated by cell sorting and subsequently subjected to clonal dilution to establish single-cell-derived clones for further validation of CD9 knockout efficiency. WT and CD9 KO A549 cells overexpressing human ACE2 (A549-ACE2), and 293T cells overexpressing ACE2 were obtained by transducing the cells with a retrovirus expressing human ACE2 and a blasticidin resistance gene (kindly provided by Pablo Gastaminza, National Center for Biotechnology, Madrid, Spain). These cells were grown in the presence of 2.5 ug/mL of blasticidin (ThermoFisher Scientific, Waltham, MA, USA).

We used a SARS-CoV-2 strain isolated from Vero E6 cells, originating from a nasal swab from a patient infected in Madrid, Spain, at the beginning of 2020, termed SARS-CoV-2-MAD6 strain. This strain is similar to the B.1 variants, and it was kindly provided by Prof. Luis Enjuanes at Centro Nacional de Biotecnologia, CNB-CSIC, Spain [33]. Particularly, the SARS-CoV-2-MAD6 genome was sequenced and its sequence was identical to the SARS-CoV-2 Wuhan-Hu-1 isolate (SARS-CoV-2-WH1) genome (GenBank MN908947), with the exception of the silent mutation C3037 > T, and two missense mutations: C14408 > T (affecting nsp12) and A23403 > G (leading to D614G in S protein), as previously described [33]. The SARS-CoV-2 was grown in Vero E6 cells. SARS-CoV-2 was titrated by plaque assay (plaque-forming units, PFU/mL) in confluent monolayers of Vero E6 cells seeded in 24-well plates, as previously described [34].

Plasmids. A polymerase II expression pcDNA3.1 plasmid encoding CD9 (GenBank accession number NM_022873) was generated by synthesizing the open reading frame of CD9 (IDT DNA, Inc., Coralville, IA, USA) flanked by the unique restriction sites for *BamHI* and *NheI* and cloning this fragment in the pcDNA3.1 plasmid cut with the same enzymes.

SARS-CoV-2 infections. WT and CD9 KO SW480 and A549-ACE2 cells were seeded in 24-well plates at 2 × 10^5^ cells per well. The following day, cell monolayers, at 95% confluency, were infected with SARS-CoV-2 at a multiplicity of infection (MOI) of 0.2, in medium DMEM supplemented with 2% fetal bovine serum, and 50 mg/mL gentamicin, all from Gibco (Grand Island, NY, USA). At 24, 48, and 72 h post-infection (hpi), cell culture supernatants were collected and titrated in Vero E6 cells as previously described [34].

Infection in the presence of tetraspanin antibodies. SW480 and A549-ACE2 cells grown in 96-well plates were incubated for 30 min at 37 °C with an anti-CD9 antibody (clone M-L13, Fisher Scientific, Waltham, MA, USA) or a control antibody (mouse IgG1 kappa isotype control (P3.6.2.8.1)) at 0.12 ug/ul (approx. 10^7^ antibodies/cell), as previously described [25]. Afterwards, SARS-CoV-2 was added for 2 h at 37 °C, and then the cells were rinsed and incubated at 37 °C for 48 h. Viral titers at 48 hpi were measured by a plaque lysis assay in Vero E6 cells, as previously described [34].

Silencing of NRP1. Human A549-ACE2 cells were transfected with a small interfering RNA specific for human NRP1 (ThermoFisher Scientific, Waltham, MA, USA, s16843) or with the non-targeting (NT) negative control (ThermoFisher Scientific, Waltham, MA, USA, AM4635), twice, 24 h apart. The siRNAs were transfected at a final concentration of 20 nM, using lipofectamine RNAiMax (ThermoFisher Scientific, Waltham, MA, USA), according to the manufacturer’s instructions.

qRT-PCRs. The mRNA levels of *ACE2, NRP1, furin, TMPRSS2,* and *β-actin* were analyzed by RT-qPCR. To this end, total RNAs from cells were extracted using the total RNA extraction kit (Omega Biotek, Norcross, GA, USA). The High-Capacity cDNA transcription kit (ThermoFisher Scientific, Waltham, MA, USA) was used for performing the reverse transcriptase (RT) reactions at 37 °C, for 2 h, using random hexamers and total RNA as template. qPCRs were performed using the cDNAs obtained from the RT reactions, the Power SybrGreen PCR master mix (ThermoFisher Scientific, Waltham, MA, USA), and primers specific for human *ACE2* (Fw: 5′-CAAGAGCAAACGGTTGAACAC-3′ and Rv: 5′-CCAGAGCCTCTCATTGTAGTCT-3′), *NRP1* (Fw: 5′-TTCAGGATCACACAGGAGATGG-3′ and Rv: 5′-TAAACCACAGGGCTCACCAG-3′), *furin* (Fw: 5′-AGATGGGTTTAATGACTGG-3′ and Rv: 5′-CATAGAGTACGAGGGTGAAC-3′), *TMPRSS2* (Fw: 5′-GATGACAGCGGATCCACCAG-3′ and Rv: 5′-TTGACCCCGCAGGCTATACA-3′), and *β-actin* (Fw: 5′-GCATCCTCACCCTGAAGTA-3′ and Rv: 5′-CACGCAGCTCATTGTAGAAG-3′) genes. Quantification was achieved using the threshold cycle (2^−ΔΔCT^) method [35] and normalized to actin expression levels.

Western blots. Cells were lysed in lysis buffer (0.5% NP-40, 50 mM Tris HCl pH 7.5, 250 mM NaCl, 1 mM EDTA) supplemented with protease (ThermoFisher Scientific, Waltham, MA, USA) and phosphatase (Merck, Darmstadt, Germany) inhibitors. Then, cell lysates were mixed with Laemmli sample buffer (Biorad, Hercules, CA, USA), containing β-mercaptoethanol (Sigma Aldrich, Saint Louis, MO, USA, M3148), and heated at 95 °C for 5 min, before SDS-PAGE electrophoresis. Proteins were transferred to nitrocellulose membranes (Biorad, Hercules, CA, USA), and detected using a rabbit monoclonal antibody (mAb) specific for CD9 (Abcam, Cambridge, UK, ab236630), rabbit mAb specific for ACE2 (Abcam, Cambridge, UK, ab108209), a rabbit mAb specific for NRP1 (Abcam, Cambridge, UK, ab81321), a mouse mAb specific for furin (Santa Cruz Biotechnology, Dallas, TX, USA, sc-133142), a mouse mAb specific for TMPRSS2 (Santa Cruz Biotechnology, Dallas, TX, USA, sc-515727), a mouse mAb specific for actin (Sigma Aldrich, Saint Louis, MO, USA, A1978), and a mouse mAb specific for flotillin (BD Biosciences, San Jose, CA, USA, 610820), followed by binding to goat anti-rabbit or goat anti-mouse polyclonal antibodies conjugated to horseradish peroxidase (Sigma Aldrich, Saint Louis, MO, USA, 12-349). Nitrocellulose membranes were then revealed by chemiluminescence with the SuperSignal west femto maximum sensitivity substrate (Thermo Fisher Scientific, Waltham, MA, USA). Where indicated, protein bands were quantified by densitometry using the ImageJ (Fiji) software (version 1.54f, National Institute of Health, Bethesda, MD, USA).

Flow cytometry. A549 and SW480 cells, WT and CD9 KO cells were detached using TrypLE™ Express Enzyme (Gibco, Waltham, NY, USA) at 37 °C for 5 min, collected, and washed twice with cold phosphate-buffered saline (PBS) supplemented with 2% fetal bovine serum and 2 mM EDTA. Approximately 1 × 10^6^ cells per sample were incubated with an APC-conjugated anti-CD9 antibody (BD Biosciences, San Jose, CA, USA, Clone M-L13, 341638) for 30 min at 4 °C in the dark. After staining, cells were washed with PBS containing 2% FBS and 2 mM EDTA and resuspended in the same buffer for analysis. Fluorescence was measured using a Gallios flow cytometer (Beckman Coulter, Brea, CA, USA), and data were analyzed using FlowJo software (Ashland, OR, USA, version 10.8.1).

Immunofluorescence. Confluent monolayers of human SW480 and A549 cells were grown on sterile glass coverslips on 24-well plates until 90% confluency. The cells were fixed and permeabilized with 10% formaldehyde and 0.1% triton X100 for 10 min at RT. Then, the cells were blocked with PBS containing 2.5% bovine serum albumin during 1 h at RT, and the cells were incubated with the following antibodies: anti-CD9 (generated in rabbit, ab236630, Abcam, Cambridge, UK), and anti-ACE2 (generated in mouse, sc-390851), anti-NRP1 (generated in mouse, sc-5307), anti-furin (generated in mouse, sc133142), anti-TMPRSS2 (generated in mouse, sc-515727), and anti-flotillin (generated in mouse, sc-7465), all of them from Santa Cruz Biotechnology, Dallas, TX, USA, at 4 °C overnight. Coverslips were washed 4 times with PBS and stained with secondary anti-rabbit and anti-mouse antibodies conjugated with Alexa Fluor 594 and 647 (Thermo Fischer Scientific, Waltham, MA, USA), respectively, for 1 h at RT, and nuclei were stained with DAPI (Thermo Fisher Scientific) for 20 min at RT. Coverslips were washed 4 times with PBS, and mounted in ProLong Gold antifade reagent (Thermo Fisher Scientific, Waltham, MA, USA) and analyzed on a Leica STELLARIS 5 confocal microscope (Leica Microsystems, Wetzlar, Germany). Images were acquired with the same instrument setting and analyzed using the Fiji software (version 1.54f, National Institute of Health, Bethesda, MD, USA).

Isolation of detergent-resistant membranes. A protocol adapted from [36] was followed. For each preparation of SW480 and A549-ACE2 WT and CD9 KO cells, cells from four 10 cm culture plates were washed three times with phosphate-buffered saline (PBS), scraped off in PBS supplemented with protease inhibitors, and recovered by centrifugation. Sedimented cells were resuspended in 1 mL Raft buffer (20 mM HEPES NaOH, pH 7.4, 150 mM NaCl, 1 mM EDTA) containing 1% Brij98 detergent (Thermo Fisher Scientific, Waltham, MA, USA) and protease inhibitors, by five passages through a 25 G cannula. After determining the protein concentration with the Pierce BCA Protein Assay Kit (Thermo Fisher Scientific, Waltham, MA, USA), equal protein amounts of each sample were adjusted to a total volume of 1 mL with Raft buffer containing a final concentration of 1% Brij98. Then, samples were kept at 37 °C for 5 min, followed by a 30 min incubation on ice. The lysates (1 mL) were adjusted to a sucrose concentration of 43% by mixing with 2 mL 65% (*w/v*) sucrose in Raft buffer. To construct a discontinuous sucrose gradient, this sample was layered beneath 7 mL 35% (*w/v*) and 3 mL 5% (*w/v*) sucrose in Raft buffer, respectively. Separation was performed in an SW40Ti rotor (Beckman Coulter, Brea, CA, USA) at 263,627 x gmax for 16 h at 4 °C. Fractions of 1 mL were collected from top to bottom, for a total of 12 fractions. Aliquots of each fraction were supplemented with the required amount of 5× SDS sample buffer and analyzed by Western blotting.

DuoLink^®^ Proximity ligation assays (PLA). SW480 and A549-ACE2 WT and CD9 KO cells were seeded in Ibidi 8-well µ-slides (Ibidi, Gräfelfing, Germany, cat. no. 80826) and fixed with 4% paraformaldehyde (PFA) for 10 min at room temperature. The PLA assay was subsequently performed using the Duolink^®^ In Situ Red Starter Kit Mouse/Rabbit (DUO92101, Sigma-Aldrich, Saint Louis, MO, USA), following the manufacturer’s instructions. Briefly, after fixation, cells were incubated with the blocking solution provided in the kit for 1 h at 37 °C. Then, cells were incubated overnight at 4 °C with primary antibodies specific for CD9 (rabbit, ab236630, Abcam, Cambridge, UK) in combination with either ICAM1 (mouse, clone 6.5B5, sc-18853) as a positive control, or antibodies against ACE2 (mouse, sc-390851), NRP1 (mouse, sc-5307), furin (mouse, sc-133142), or TMPRSS2 (mouse, sc-515727), all from Santa Cruz Biotechnology (Dallas, TX, USA). The next day, after three PBS washes, Duolink^®^ PLA Probes Anti-Rabbit PLUS and Anti-Mouse MINUS were applied at a 1:5 dilution in antibody diluent for 1 h at 37 °C. Slides were washed with Duolink In Situ Wash Buffer A, then incubated with ligation mixture (ligation buffer and ligase) for 30 min at 37 °C, facilitating connector oligonucleotide circularization if the probes were within <40 nm proximity. After additional washes, this was followed by an amplification step using amplification buffer plus polymerase that was added for a 100 min incubation at 37 °C, yielding rolling circle amplification of each ligation product with red fluorophore incorporation. The slides were washed twice with Wash Buffer B (10 min each), followed by a final wash with 0.01× Wash Buffer B. Nuclei were stained using Duolink^®^ In Situ Mounting Medium with DAPI provided in the kit. Hybridization of complementary labeled oligonucleotides to the amplified product produced discrete red fluorescent signals (dots). Images were acquired using a fluorescence microscope equipped with an ApoTome.2 optical sectioning module (Axio Imager.Z2, Carl Zeiss Microscopy GmbH, Jena, Germany) using a Texas Red filter set (λ_ex = 594 nm; λ_em = 624 nm). Quantification of the number of red dots per cell was performed using Fiji (ImageJ, version 1.54f, National Institute of Health, Bethesda, MD, USA).

Co-immunoprecipitation assays. 293T-ACE2 cells (100 mm plate format) were transfected with a plasmid expressing CD9 or the empty plasmid, using lipofectamine 3000 (Thermo Fisher Scientific, Waltham, MA, USA), according to the manufacturer’s instructions. At 24 h post-transfection, the cells were lysed in the co-immunoprecipitation buffer (NaCl 250 mM; EDTA 1 mM; 50 mM TrisHCl, pH 7.5; Brij98 0.5%) containing protease (ThermoFisher Scientific, Waltham, MA, USA) and phosphatase (Merck, Rahway, NJ, USA) inhibitors, and cleared by centrifugation. Alternatively, A549-ACE2 WT and A549-ACE2 CD9 KO cells (100 mm plate format) were lysed in the co-immunoprecipitation buffer containing protease and phosphatase (Merck, Rahway, NJ, USA) inhibitors, and cleared by centrifugation. Protein G-dynabeads (Thermo Fisher Scientific, Waltham, MA, USA, 1004D) were incubated with the anti-CD9 antibody (Thermo Fisher Scientific, Waltham, MA, USA, 1004D) overnight at 4 °C, and washed three times with Tris-buffered saline (TBS). Then, the anti-CD9 antibody conjugated to protein G-dynabeads was incubated with the cleared cell lysates for 4 h at room temperature. The cellular extracts combined with the affinity resins were washed five times in TBS buffer containing 0.05% Tween-20. The immunoprecipitated proteins were unbound using 0.1 M glycine buffer at pH 2.4, denatured in loading buffer, and incubated at 95 °C for 5 min. Then, samples were analyzed by electrophoresis and Western blot as described above.

## 3. Results

The tetraspanin CD9 facilitates SARS-CoV-2 infection. Tetraspanins facilitate the entry of many viruses, including hepatitis C virus [37], human papillomavirus [28], IAV [25,31], and coronaviruses, including HCoV-229E, MERS-CoV, and SARS-CoV [25,26]. To analyze whether the tetraspanin CD9 facilitates the infection by SARS-CoV-2, the colorectal SW480/WT and SW480/CD9 KO cells, and the lung adenocarcinoma A549/WT and A549/CD9 KO cells, were generated using a home-made CRISPR/Cas9 system (see Materials and Methods section). First, we confirmed by Western blot (Figure 1A), flow cytometry (Figure 1B), and immunofluorescence (Figure 1C), using antibodies specific for CD9, that the KO cells did not express the CD9 protein. The A549/WT and A549/CD9 KO cells were additionally transduced to overexpress human ACE2, given that A549 cells, which do not express ACE2, do not get efficiently infected [38]. Then, the SW480/WT and SW480/CD9 KO cells were infected, and viral titers were determined at 24, 48, and 72 hpi. At 48 and 72 hpi, 3- and 5-fold reductions in viral titers were detected in SW480/CD9 KO cells, compared to SW480/WT cells, respectively (Figure 1D), suggesting that CD9 facilitates SARS-CoV-2 infection. To analyze whether the effect of CD9 also applies to other cell lines, similar experiments were performed in A549-ACE2/WT and A549-ACE2/CD9 KO cells, showing 5 and 4-fold reductions in viral titers in the A549-ACE2/CD9 KO cells, compared to the A549-ACE2/WT cells (Figure 1D), further confirming that CD9 facilitates SARS-CoV-2 infection. Furthermore, to reinforce these results, SW480 and A549-ACE2 cells were incubated with an anti-CD9 blocking antibody, or with an IgG isotype control, being the anti-CD9 antibody concentration chosen based on a previous report in which the authors use the same commercial anti-CD9 antibody, and they show that blocking the cells with this antibody concentration reduces the infectivity of the coronavirus MHV, of an influenza virus, and of Vesicular Stomatitis virus (VSV) particles pseudotyped with the VSV G protein, and with the spike proteins from the coronaviruses HCoV-229E, MERS-CoV, and SARS-CoV [25]. SARS-CoV-2 titers were measured at 48 hpi, showing reduction of viral titers in the cells treated with the anti-CD9 antibody, compared to the cells treated with the isotype antibody control (Figure 1E).

Blocking CD9 expression decreases NRP1 protein levels. Given the results showing that ACE2 is essential for SARS-CoV-2 entry into the cells, that furin, TMPRSS2 and NRP1 facilitate SARS-CoV-2 cellular entry, that tetraspanin-enriched microdomains (TEMs) are known to contain a variety of cell-surface proteases [36,39], and that TEMs may contain coronavirus receptors and priming proteases [25,26], we hypothesized that CD9 could be affecting the expression or the membrane localization of ACE2, furin, TMPRSS2, and NRP1. To this end, we performed Western blot assays using antibodies specific for these proteins in SW480/WT and SW480/CD9 KO cells, either mock infected or infected with SARS-CoV-2 for 24, 48, and 72 h (Figure 2A). Of note, the expression of NRP1 diminished both in mock-infected and SARS-CoV-2-infected SW480/CD9 KO cells compared to mock-infected and SARS-CoV-2-infected SW480/WT cells (Figure 2A), suggesting that CD9 may affect the expression or stability of NRP1. On the other hand, CD9 KO cells exhibit a slight increase in TMPRSS2, showing sustained elevation across all time points evaluated (Figure 2A), highlighting a regulatory role for CD9 in controlling the expression of SARS-CoV-2 entry factors. To further analyze whether CD9 expression increases NRP1 protein levels, the levels of NRP1 were also analyzed in A549-ACE2/WT and A549-ACE2/CD9 KO cells, showing a reduction in the NRP1 levels in the A549-ACE2/CD9 KO cells compared to the A549-ACE2/WT cells in SARS-CoV-2-infected cells (Appendix A). To analyze whether CD9 might affect the expression of *NRP1* and the other SARS-CoV-2 entry molecules at the transcript level, the levels of mRNA were analyzed by RT-qPCR in both WT and CD9 KO SW480 cell lines, either mock-infected or infected with SARS-CoV-2 for 24, 48, and 72 h. No significant differences at the mRNA levels were observed between SW480/WT and SW480/CD9 KO cells, either in mock-infected conditions or in SARS-CoV-2 conditions (Figure 2B), suggesting that CD9 does not affect the transcription levels of *NRP1* or any of the other entry factors.

Given that it had been shown that NRP1, which is known to bind furin-cleaved substrates, potentiating SARS-CoV-2 infectivity [17,18], we analyzed in our cell system whether the decreased expression of NRP1 decreased viral infection. To this end, A549-ACE2 cells were transfected with an siRNA specific for NRP1 and with a non-targeted siRNA as control. The efficient knockdown of NRP1 was confirmed by Western blot (Figure 3A). Interestingly, in cells silenced for NRP1, SARS-CoV-2 titers decreased by around 10-fold at 48 and 72 hpi, in comparison with the control cells transfected with non-targeted siRNA (Figure 3B). These data correlate with previous data showing that NRP1 facilitates SARS-CoV-2 entry and infectivity, as previously described [17,18]. In addition, these data provide a molecular mechanism for the effect of CD9 on facilitating SARS-CoV-2 infection.

CD9-enriched membranes contain the SARS-CoV-2 receptor ACE2 and S-cleavage proteases. Previous results suggested that tetraspanin- and particularly CD9-enriched microdomains contained the coronavirus receptors APN, DPP4, CEACAM, and ACE2; and the protease TMPRSS2 [25], suggesting that CD9 might facilitate SARS-CoV-2 infection by bringing in proximity the SARS-CoV-2 receptor and the proteases that facilitate the viral entrance into the cellular plasmatic membrane. To this end, to analyze whether CD9, ACE2, NRP1, and the proteases furin and TMPRSS2 are located in the same membrane microdomains, a fractionation experiment using the Brij98 detergent was performed in a similar way as previously reported [36]. Lysates from SW480/WT and SW480/CD9 KO cells were layered beneath a discontinuous sucrose step gradient, from which fractions were recovered from top to bottom after overnight ultracentrifugation for Western blot analysis, and 12 fractions were collected. Interestingly, CD9, ACE2, NRP1, and the transmembrane proteases TMPRSS2 and furin were mainly detected in the same fractions, which were the detergent-resistant membrane (DRM)-fractions, as observed by Western blot analysis using antibodies specific for each of these proteins (Figure 4A), suggesting that these proteins are located to the same membrane microdomains. As a control, we visualized the distribution of the established lipid raft marker flotillin-1 [40,41], which was also mainly present in the DRM fractions (Figure 4A), as previously reported [36]. To analyze whether CD9 affects the location or the expression levels of ACE2, NRP1, and the proteases TMPRSS2 and furin, the presence of these proteins was compared among SW480/WT and SW480/CD9 KO cells. Western blot analysis revealed that the presence of ACE2, NRP1, and the proteases TMPRSS2 and furin was significantly lower in the DRM fractions containing CD9 (as observed in the WT cells), in the SW480/CD9 KO cells compared to the SW480/WT cells (Figure 4B). To reinforce these results, the same fractionation studies were performed in the A549-ACE2/WT and A549-ACE2/CD9 KO cells (Appendix A), showing that CD9 was mainly located at the same fractions as ACE2, NRP1, furin, and TMPRSS2. However, in this case, whereas the proteins ACE2, NRP1, furin, TMPRSS2, CD9, and flotillin were preferentially expressed at fraction 5, in the A549-ACE2/WT cells, the proteins ACE2, NRP1, and furin were preferentially visualized at fraction 3 in the A549-ACE2/CD9 KO cells, and TMPRSS2 could not be detected (Appendix A). Furthermore, the amounts of ACE2, NRP1, furin, and TMPRSS2 were significantly lower in the A549-ACE2/CD9 KO cells, in comparison to the A549-ACE2/WT cells (Appendix A). These results strongly suggest that CD9 affects the protein levels of ACE2, NRP1, and the proteases TMPRSS2 and furin within the cellular membranes’ microdomains, and suggest that CD9 likely helps to bring together the host factors ACE2, NRP1, and the proteases TMPRSS2 and furin, facilitating viral infectivity.

CD9 partially colocalizes with ACE2, NRP1, and the proteases TMPRSS2 and furin. To reinforce the results showing that CD9 likely facilitates the proximity of ACE2, NRP1, and the proteases TMPRSS2 and furin, immunofluorescence and confocal studies were assessed. To this end, SW480 cells were stained with antibodies specific for CD9, together with antibodies specific for ACE2, NRP1, TMPRSS2, furin, and flotillin, as control (Figure 5). A partial colocalization in the peri-plasmatic membrane region between CD9 and ACE2, NRP1, TMPRSS2, furin, and flotillin was observed, suggesting that these proteins might be in the same membrane microdomains (Figure 5).

Furthermore, a proximity ligation assay (PLA) was employed to investigate protein-protein interactions between CD9 and the proteins of interest, including NRP1, furin, ACE2, and TMPRSS2. ICAM-1 was used as a positive control, as its interaction is well documented [42], and serves to validate the assay’s efficiency (Figure 6A). To this end, SW480 and A549-ACE2/WT (Figure 6) and A549-ACE2/CD9 KO cells (Appendix A) were fixed and incubated with primary antibodies specific for CD9 together with antibodies specific for ICAM1 (as a positive control) (Figure 6B) or against ACE2, NRP1, furin or TMPRSS2 (Figure 6C-D), followed by an incubation with secondary antibodies conjugated to proximity ligation assay (PLA) probes. Then, an incubation with a solution containing fluorescently labeled nucleotides, complementary oligonucleotides, and a polymerase was applied. Finally, the cells were incubated with a ligation enzyme. The results revealed a significant proximity between CD9 and both NRP1 and furin, as evidenced by the presence of PLA signal dots in both SW480/WT (Figure 6C) and A549-ACE2/WT cells (Figure 6D). As control, no signal dots were observed for SW480/CD9 KO nor for A549-ACE2/CD9 KO cells (Appendix A), indicating proximity between CD9 and NRP1 and furin proteins (30–40 nm). In addition, we could detect that CD9 is in close proximity to TMPRSS2 and ACE2 in the A549-ACE2/WT cells (Figure 6D), but not in the SW480/WT cells (Figure 6C), likely due to the fact that the expression of TMPRSS2 and ACE2 is lower in the SW480 cells compared to the A549-ACE2 cells These findings suggest that CD9 participates in molecular interactions with NRP1, furin, and, to a lesser extent, with TMPRSS2 and ACE2, potentially playing a role in the regulation or modulation of their functions, possibly through membrane organization or the mediation of specific protein complexes.

CD9 interacts with ACE2. To analyze whether CD9 interacts with NRP1, furin, ACE2, and TMPRSS2, as strongly suggested by the Duolink assays, cellular extracts from A549-ACE2/WT cells and A549-ACE2/CD9 KO cells (as control) were immunoprecipitated with an antibody specific for CD9 conjugated to protein G-dynabeads. Alternatively, 293T-ACE2 cells were transfected with a pcDNA3.1 plasmid expressing CD9 or with the empty plasmid, as control, and the cellular extracts were immunoprecipitated with the anti-CD9 antibody conjugated to protein G-dynabeads. First, we analyzed CD9 and ACE2 protein expression in the cellular extracts by Western blot (Figure 7A). Interestingly, CD9 and ACE2 co-immunoprecipitated together, both in the 293T-ACE2 cells transfected with the plasmid overexpressing CD9 or in the A549-ACE2/WT cells (Figure 7B), indicating that these two proteins interact directly or indirectly. As control, ACE2 was not detected after the co-immunoprecipitation using the A549-ACE2/CD9 KO cells (Figure 7B). However, we were not able to detect NRP1, TMPRSS2, and furin after immunoprecipitating CD9. This could be due to a low sensitivity detection of these proteins (Appendix A), because all the proteins ACE2, NRP1, TMPRSS2, and furin are detected as being in close proximity to CD9 according to the Duolink assays (Figure 6), and because NRP1, TMPRSS2, and furin are detected to lower levels than ACE2 by Western blot (Appendix A).

## 4. Discussion

In this manuscript, we show that CD9 facilitates SARS-CoV-2 infection. In addition, we show how knocking out CD9 leads to a decrease in the protein expression of NRP1, a protein that improves SARS-CoV-2 infection. Furthermore, we show that CD9 colocalizes with ACE2, NRP1, furin, and TMPRSS2 at the plasma membrane; that the absence of CD9 decreases the expression of these proteins on the CD9-enriched microdomains, and that CD9 interacts with ACE2, suggesting that CD9 facilitates SARS-CoV-2 infection and that CD9 brings together different host proteins involved in SARS-CoV-2 attachment and entry into host cells, such as ACE2, NRP1, furin, and TMPRSS2.

Using two different cell lines of CD9 knock-out cells, and the treatment of WT cells with an antibody blocking CD9, we have shown that SARS-CoV-2 titers are decreased. Similarly, it was previously shown that the treatment of cells with antibodies specific for the Tspans CD9, CD63, and CD81 reduced the infection by the coronavirus MHV and SARS-CoV, MERS-CoV, and HCoV-229E pseudovirus transductions in susceptible cells [25]. Interestingly, antibody treatment of tetraspanins did not reduce the levels of CoV binding to the entry receptor, and MHV infection and pseudovirus transductions were rescued with overexpression of TMPRSS2, thus suggesting that tetraspanin facilitation of CoV infections lies in mechanically allowing the access of receptor-bound viruses to the transmembrane proteases that speed up membrane fusion [25]. Importantly, the fact that a blocking antibody targeting CD9 can effectively reduce SARS-CoV-2 titers highlights not only the mechanistic role of CD9 in viral entry but also offers translational potential, suggesting that tetraspanin-targeting antibodies could be developed as therapeutic agents against SARS-CoV-2 and possibly other coronaviruses, with meaningful implications for clinical intervention.

Our Western blot analysis reveals that knockout of CD9 in both SW480 and A549 cells leads to a marked reduction in NRP1 expression compared to WT cells following SARS-CoV-2 infection. In contrast, the expression of the protease TMPRSS2 was increased in SW480/CD9 KO cells, with TMPRSS2 elevated across all time points examined. Similarly, the expression of TMPRSS2 was slightly decreased in the A549-ACE2/CD9 KO cells. The observed reduction in NRP1 levels in CD9-deficient cells suggests that CD9 may play a role in stabilizing or maintaining the expression of this key SARS-CoV-2 entry factor. CD9, a member of the tetraspanin family, is known to organize membrane microdomains and facilitate the assembly of protein complexes at the cell surface [43]. Loss of CD9 could disrupt these microdomains, potentially leading to increased internalization, degradation, or reduced trafficking of NRP1 to the plasma membrane. TMPRSS2 is essential for spike protein activation and viral membrane fusion [38]. TMPRSS2 can also cleave ACE2, thereby reducing its cell-surface expression and promoting its shedding [44]. Additionally, CD9 is known to negatively regulate the activity of the metalloproteinase ADAM17 [45,46], which is involved in ectodomain shedding of ACE2 [44]. Therefore, the CD9-mediated cell entry of SARS-CoV-2 could dysregulate the shedding of the ACE2 receptor by ADAMT17 [47]. Loss of CD9 may therefore lead to increased ADAM17 activity and further ACE2 cleavage, contributing to lower steady-state ACE2 levels in CD9 KO cells at the CD9-enriched microdomains. However, a proteolytic cleavage for NRP1 has not been described for furin nor TMPRSS2, nor ADAM17 or any other metalloproteinase regulated by CD9 [48]. Further studies will be required to assess whether CD9 differentially regulates the proteolytic processing of the diverse array of cell-surface receptors involved in SARS-CoV-2 entry.

We have shown that CD9 is in the same plasma membrane microdomains as ACE2, NRP1, furin, and TMPRSS2 by using gradient ultracentrifugation. Similar to our results for TMPRSS2, it was previously shown that the tetraspanin CD9, but not the tetraspanin CD81, formed cell-surface complexes of dipeptidyl peptidase 4 (DPP4), the MERS-CoV receptor, and the type II transmembrane serine protease (TTSP) member TMPRSS2, a CoV-activating protease [26]. Furthermore, it was previously found that the CoV receptors ACE2, APN, CEACAM, and DPP4 and the protease TMPRSS2 are more abundant on TEMs than elsewhere in cell surfaces [25]. However, in this manuscript, the authors only distinguish between high-density and low-density fractions, and as far as we know, it was not previously described that CD9 is in the same cell-surface microdomains as NRP1 and furin.

There is evidence that cathepsins are located at endosomes, and there is no evidence that cathepsins are located at TEMs; therefore, the effect of CD9 on SARS-CoV-2 entry likely applies to plasma membrane-mediated SARS-CoV-2 entry, and not to the entry via endosomes (reviewed in [16]). It seems that the SARS-CoV-2 entry at the plasma membrane via TMPRSS2 is faster than the endosomal entry route, and the first route is the preferred entry route when TMPRSS2 is expressed in the cells [14]. However, analyses of clinical SARS-CoV-2 isolates for their entry into cells reflecting in vivo infection environments, taking into account the cells in which CD9 is expressed, may be necessary to assess the importance of TEM-associated S proteolysis in natural infection and disease.

In addition, we have found by immunoprecipitation and Western blot that ACE2 co-immunoprecipitates with CD9. However, we have not been able to detect NRP1, TMPRSS2, and furin after immunoprecipitating CD9. This could be due to a low sensitivity detection of these proteins, because all the proteins ACE2, NRP1, TMPRSS2, and furin are detected in close proximity to CD9 according to the Duolink assays, and because NRP1, TMPRSS2, and furin are detected at lower levels than ACE2. According to our results, it was previously shown that ACE2 co-immunoprecipitates with CD9 [49]. While co-IP and Western blotting confirmed a stable interaction between ACE2 and CD9, Duolink revealed more frequent or abundant proximity events between CD9 and NRP1 or furin, and ACE2 interaction was only detected in the A549-ACE2 cell line, where ACE2 is overexpressed, but not in the SW480 cell line. Moreover, the relatively weaker Duolink signal for ACE2–CD9, despite co-IP detection, is surprising. These could be due to the different affinities of the antibodies used, which could work better for one technique versus the other, but underscore the value of Duolink in uncovering interactions that are otherwise missed by traditional biochemical methods, such as the case for furin or NRP1, providing a more nuanced view of the interactome in its physiological context. A key advantage of the Duolink proximity ligation assay (PLA) over traditional co-immunoprecipitation (Co-IP) is its ability to detect not only stable, but also transient and low-affinity protein-protein interactions directly in situ. Unlike Co-IP, which requires cell lysis and multiple washing steps—conditions that can disrupt labile or weakly associated complexes—Duolink is performed on fixed cells or tissue. This fixation step effectively “freezes” the molecular state of the cell at a specific time point, preserving even fleeting or low-abundance interactions that may be lost during biochemical extraction [50].

In summary, our findings demonstrate that CD9 plays a central role in organizing the SARS-CoV-2 entry platform by facilitating the colocalization and functional interplay of ACE2, NRP1, furin, and TMPRSS2 within plasma membrane microdomains. The reduction in viral titers upon CD9 knockout or blockade, together with the altered expression and distribution of entry factors, underscores the importance of tetraspanin-mediated receptor-protease clustering for efficient SARS-CoV-2 infection. These results highlight the potential of targeting CD9 as a therapeutic strategy to disrupt viral entry and provide new insights into the molecular mechanisms governing coronavirus–host interactions.

## Figures and Tables

**Figure 1 viruses-17-01141-f001:**
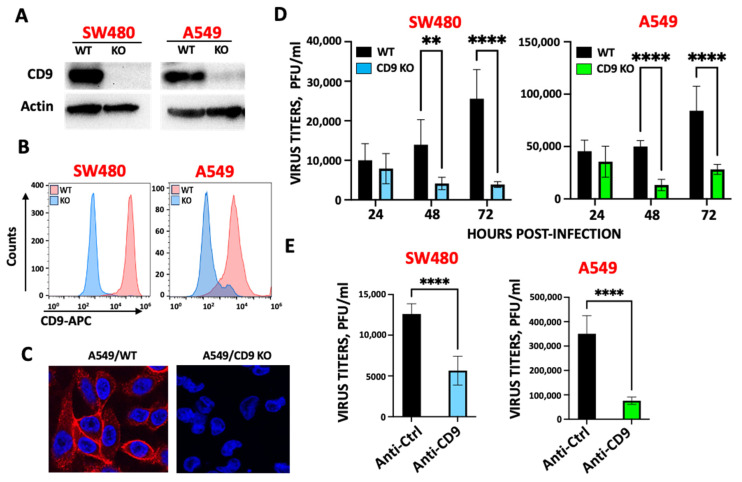
The protein CD9 facilitates SARS-CoV-2 infection. SW480 and A549 WT and CD9 KO cells, previously generated, were used. (**A**) Cellular extracts from SW480 and A549 WT and CD9 KO cells were obtained, and Western blots with antibodies specific for CD9 and actin as control were performed. (**B**) The expression of CD9 was analyzed by flow cytometry in both SW480 and A549 WT and CD9 KO cells, using an anti-CD9-APC antibody. (**C**) The expression of CD9 was analyzed by immunodetection followed by fluorescence microscopy. To this end, A549/WT and A549/CD9 KO cells were fixed and permeabilized, and the cells were stained with an anti-CD9 antibody (in red). DAPI was used for nuclear staining (in blue). Representative images are included. (**D**) SW480 and A549-ACE-2 WT and CD9 KO cells were infected with SARS-CoV-2, and supernatants were collected at 24, 48, and 72 hpi. Viral titers were determined by a lysis plaque assay on Vero E6 cells. (**E**) SW480 and A549 WT cells were treated either with an antibody blocking the CD9 protein or with an isotype control antibody. Then, the cells were SARS-CoV-2 infected, and viral titers at 48 hpi were determined by a lysis plaque assay on Vero E6 cells. (**D**,**E**) Error bars represent the means and standard deviations (SD) of results from three independent experiments, including triplicate wells in each experiment. Ns, *p* > 0.05, * *p* < 0.05, ** *p* < 0.01, *** *p* < 0.001, **** *p* < 0.0001 (for comparison between WT and CD9 KO cells, in (**D**), and between cells treated with the control and the anti-CD9 antibodies, in (**E**), using an unpaired two-tailed Student’s *t*-test).

**Figure 2 viruses-17-01141-f002:**
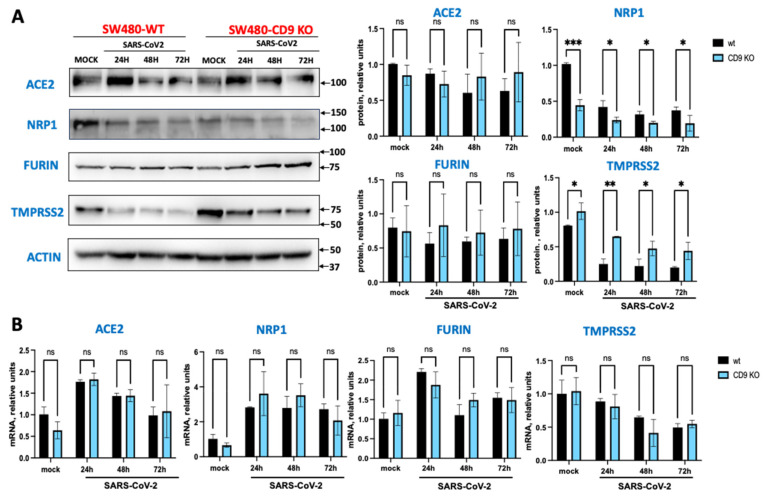
CD9 expression affects NRP1 expression levels. SW480/WT and SW480/CD9 KO cells were left mock infected or infected for 24, 48, and 72 hpi. (**A**) Protein extracts were obtained, and the levels of NRP1, ACE2, furin, TMPRSS2, and actin as control were analyzed by Western blot using specific antibodies. Two experiments were performed showing similar results. Representative blots from one out of two experiments are shown. Molecular weights (in kilodaltons) are indicated on the right. Western blots were quantified by densitometry using ImageJ software and normalized to the levels of actin in each sample (graphs on the right). For quantifications, the means and standard deviations of the two Western blots performed are represented. Ns, *p* > 0.05, * *p* < 0.05, ** *p* < 0.01, *** *p* < 0.001, (for comparison between WT and CD9 KO cells using an unpaired two-tailed Student’s *t*-test). (**B**) Total RNAs from mock-infected or SARS-CoV-2-infected SW480 cells were purified, and the expression level of *ACE2*, *NRP1*, *furin*, and *TMPRSS2* mRNAs was quantified by reverse transcriptase (RT) reaction followed by qPCR, and quantified relative to the levels of *β-actin* mRNA. Error bars represent the means and standard deviations (SD) of results from three independent experiments, including triplicate wells in each experiment. Ns, *p* > 0.05 (for comparison between WT and CD9 KO cells using an unpaired two-tailed Student’s *t*-test).

**Figure 3 viruses-17-01141-f003:**
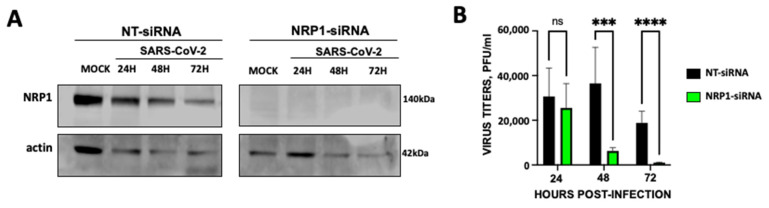
NRP1 facilitates SARS-CoV-2 infection. A549-ACE2 WT cells were silenced with an siRNA specific for NRP1 or with a non-targeted (NT) siRNA as control. (**A**) Silencing efficiency was confirmed by Western blot in mock infected and SARS-CoV-2-infected cells after 24, 48, and 72 h post-infection, using an antibody specific for NRP1 and an antibody specific for actin as control. Two experiments were performed, showing similar results. (**B**) Human A549-ACE2 cells were transfected with a non-targeted (NT) or an NRP1-specific siRNA. At 24 h post-transfection, the cells were infected with SARS-CoV-2. Cell culture supernatants were collected at 24, 48, and 72 hpi and titrated by a lysis plaque assay on Vero E6 cells. Error bars represent the means and standard deviations (SD) of results from three independent experiments, including triplicate wells in each experiment. Ns, *p* > 0.05, * *p* < 0.05, ** *p* < 0.01, *** *p* < 0.001, **** *p* < 0.0001 (for comparison between cells transfected with the NT siRNA and the cells transfected with the NRP1 siRNA, using an unpaired two-tailed Student’s *t*-test).

**Figure 4 viruses-17-01141-f004:**
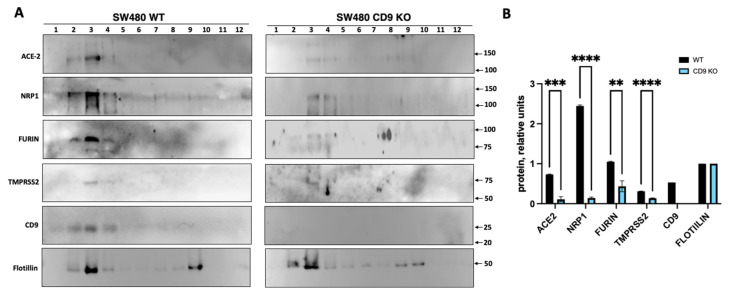
CD9 localizes to the same transmembrane domains as ACE2, NRP1, furin, and TMPRSS2, and increases the expression levels of these proteins. To analyze whether CD9, ACE-2, NRP1, and the proteases are located in the same membrane microdomains, a fractionation experiment using the Brij98 detergent was performed. Lysates from SW480/WT and SW480/CD9 KO cells were layered beneath a discontinuous sucrose step gradient, from which 12 fractions were collected from top to bottom after overnight ultracentrifugation. (**A**) Western blot analysis, using antibodies specific for ACE-2, NRP1, furin, TMPRSS2, CD9, and flotillin as control, was performed using fractions containing the same amount of protein. The molecular weights are indicated on the right (in kilodaltons). The fraction numbers are indicated at the top. Three experiments were performed showing similar results. Representative blots from one out of three experiments are shown. (**B**) The fraction 3 of the Western blots, showing the highest protein amounts, was quantified by densitometry using ImageJ software. The amounts of ACE2, NRP1, furin, TMPRSS2, and CD9 were normalized by the amount of flotillin in fraction 3 in WT and CD9 KO cells. For quantifications, the means and standard deviations from the three Western blots performed are represented. * *p* < 0.05, ** *p* < 0.01, *** *p* < 0.001, **** *p* < 0.0001 (for comparison between WT and CD9 KO cells using an unpaired two-tailed Student’s *t*-test).

**Figure 5 viruses-17-01141-f005:**
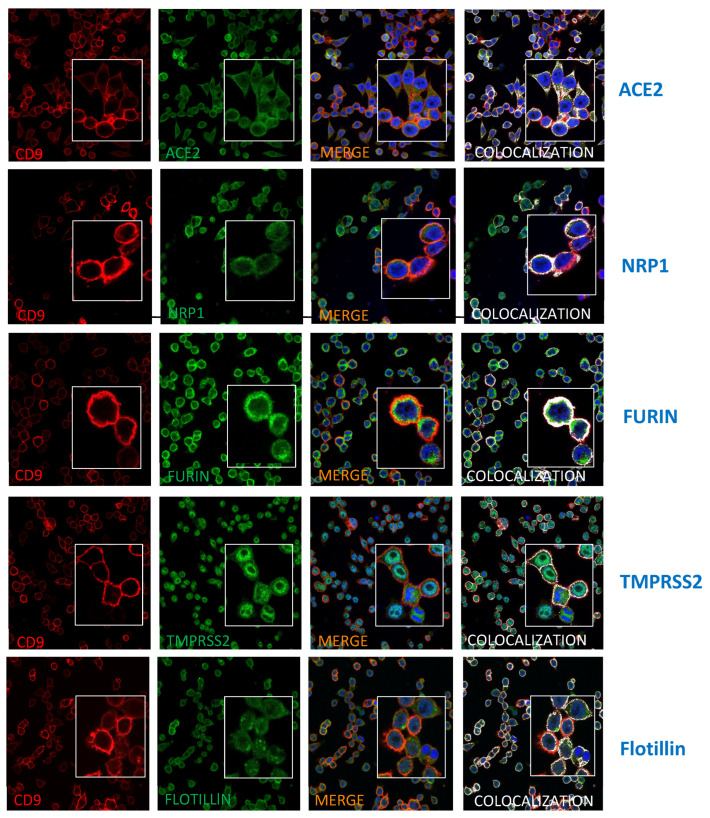
CD9 partially colocalizes with furin, TMPRSS2, NRP1, and ACE2. SW480-WT cells were fixed, permeabilized, and stained with antibodies specific for CD9, together with antibodies specific for ACE-2, NRP1, TMPRSS2, furin, and flotillin, as control. CD9 is shown in red; ACE2, NRP1, furin, TMPRSS2, and flotillin are shown in green; and nuclei were stained with DAPI and shown in blue. Areas of colocalization of both proteins appear in yellow in the third picture and in white in the fourth picture.

**Figure 6 viruses-17-01141-f006:**
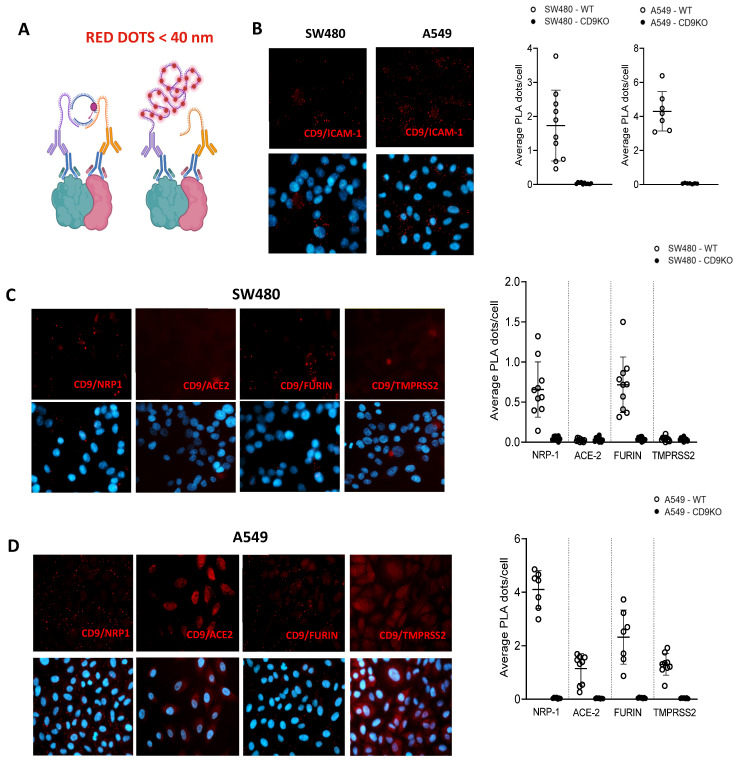
CD9 is in close proximity to NRP1 and furin. (**A**) Schematic representation of the proximity ligation assay (PLA) technique (DuoLink). PLA detects protein-protein interactions when two target proteins are within less than 40 nm of each other. Primary antibodies bind to each target protein, and species-specific secondary antibodies are conjugated to unique oligonucleotides. If the proteins are in close proximity, the oligonucleotides are ligated and amplified via rolling circle amplification, generating a fluorescent signal (visualized as red dots). (**B**) Representative images and quantification of PLA signals between CD9 and ICAM-1 in WT and CD9 KO SW480 (left) and A549-ACE2 (right) cells. (**C**–**D**) Representative PLA images showing interactions between CD9 and candidate SARS-CoV-2 entry factors (NRP1, ACE2, Furin, and TMPRSS2) in SW480/WT (**C**) and A549-ACE2/WT (**D**) cells. Nuclei are stained with DAPI (blue). Quantification of average PLA signals per cell is shown on the right of the images. Loss of PLA signal in CD9 KO cells confirms the specificity of the CD9 interactions (see Appendix A). For all images, the quantification of the average PLA signal per cell is shown to the right. For each condition, 10 images were acquired at 40× magnification, with each image containing between 80 and 120 cells.

**Figure 7 viruses-17-01141-f007:**
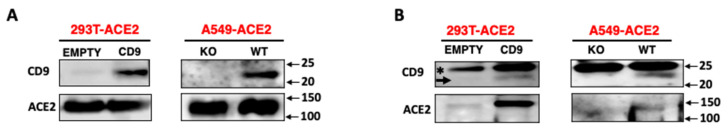
CD9 binds ACE2. (A,B) Human 293T-ACE2 cells were transiently co-transfected with a pcDNA3.1 plasmid encoding CD9 or with an empty plasmid. Alternatively, A549-ACE2/WT and A549-ACE2/CD9 KO cells were used. (**A**) The expression of CD9 and ACE2 was analyzed by Western blot in the cellular extracts. Molecular weights (in kilodaltons) are indicated on the right. (**B**) Cellular lysates were used for co-immunoprecipitation assays using an anti-CD9 antibody and dynabeads-conjugated protein G to pull down CD9. After the immunoprecipitation, CD9 and ACE2 were detected by Western blotting using antibodies specific for CD9 and ACE2. Molecular weight markers (in kilodaltons) are indicated on the right. The upper band in the anti-CD9 blot in (**B**) corresponds to the light chain of the antibody used for the co-IP (marked with an *). The band corresponding to the CD9 protein has been indicated with an arrow.

## Data Availability

The original contributions presented in this study are included in the article/Appendix A. Further inquiries can be directed to the corresponding author.

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
