# Peer review of "The Tetraspanin CD9 Facilitates SARS-CoV-2 Infection and Brings Together Different Host Proteins Involved in SARS-CoV-2 Attachment and Entry into Host Cells"

_viruses, 2025, doi:10.3390/v17081141_

Round 1

Reviewer 1 Report

Comments and Suggestions for Authors

This manuscript provides further support for the involvement of tetraspanins (particularly CD9) in entry/exit of SARS-CoV-2. However, it has a number of flwas that should be addressed prior to publication:

1) No information is provided on statistical analysis in the Methods nor in the figure legends. This must be provided.

2) Where statistical analysis appears to have been performed, significance is not explained in the figure legends.

3) In Figs 1 and 2, experimental data is from 'triplicate wells'. This would appear to be a technical replicate, where an experiment is performed only once (a single biological replicate). This is inadequate and experiments must be performed in multiple biological replicates (n = 3 is typical).

4) In Figs 3 and 6, no information is provided on replicates. This suggests only a single experiment was performed.

5) In Fig 4, data is from 3 separate experiments (n = 3 biological replicates). However, data is only shown from one experiment. This is acceptable for the Western blots but not for the densitometry data. The data from all three experiments should be combined and analysed.

6) In Fig 2, no statistical analysis is performed in panel A and yet the authors claim that there are differences in protein expression. In contrast, in panel B, mRNA data is analysed, found to be nonsignificant and stated to be not different. All data should be analysed in the same way. A similar qualitative description is also shown in Fig 6.

7) How was the antibody concentration chosen for the blocking experiment shown in Fig 1? Is it a saturating concentration, 50% saturation etc? In addition, are both the anti-CD9 and control antibodies used azide-free?

8) Previous work on tetraspanins and SARS-CoV-2 should be cited completely. There are some missing publications (Malla and Kamal, 2022 DOI: 10.2174/1381612828666220907105543 
Healy 2022 https://doi.org/10.1016/j.bbrc.2022.01.038 
Figueroa 2022 https://ecommons.luc.edu/luc_theses/4448 )

Author Response

Dear Editors and reviewer,

Please find attached the revised version of the manuscript entitled "The Tetraspanin CD9 Facilitates SARS-CoV-2 Infection and Brings Together Different Host Proteins Involved in
SARS-CoV-2 Attachment and Entry into Host Cells " by Rivero et al., which we would like to resubmit to be further considered for publication at Viruses.

We would like to thank the referees and the editors for their helpful comments, and for giving us the opportunity to revise our paper. To address the issues raised by the reviewers, we have performed additional statistical analysis and made changes to the text where appropriate. Attached you will find a complete point-by-point reply to reviewers' comments.

We think that the manuscript has been significantly improved according to the reviewer´s comments. Therefore, we hope that in the light of the changes made, you will now find our manuscript suitable for publication at Viruses.

Looking forward to hearing from you soon, sincerely,

Marta L. DeDiego

REVIEWER 1

This manuscript provides further support for the involvement of tetraspanins (particularly CD9) in entry/exit of SARS-CoV-2. However, it has a number of flwas that should be addressed prior to publication:

  • No information is provided on statistical analysis in the Methods nor in the figure legends. This must be provided.

Response: We apologize for this missing information, according to this reviewer´s comment, we have provided new statistical analysis in the legends from Figures 1, 2, 3, and 4, and supplementary Figure 2.

2) Where statistical analysis appears to have been performed, significance is not explained in the figure legends.

Response: According to this reviewer´s indication, we have explained the statistical significance in the figure´s legends.

3) In Figs 1 and 2, experimental data is from 'triplicate wells'. This would appear to be a technical replicate, where an experiment is performed only once (a single biological replicate). This is inadequate and experiments must be performed in multiple biological replicates (n = 3 is typical).

Response: Following this referee´s comment, we have specified in Figures 1D and 1E, that: “(D and E) Error bars represent standard deviations (SD) of results of measurements performed in triplicate wells. Three experiments were performed showing similar results. Ns, p>0.05, *p<0.05, **p<0.01, ***p<0.001, ****p<0.0001 (for comparison between WT and CD9 KO cells, in D, and between cells treated with the control and the anti-CD9 antibodies, in E, using an unpaired two-tailed Student´s t-test)”.

Similarly, for Figure 2, we have specified that: “For the Western blots, two experiments were performed showing similar results. Representative blots from one out of two experiments are shown. For quantifications, the mean and standard deviations, of the three Western blots performed are represented. Ns, p>0.05, *p<0.05, **p<0.01, ***p<0.001, ****p<0.0001 (for comparison between WT and CD9 KO cells using an unpaired two-tailed Student´s t-test)” and for qPCR assays, “three experiments were performed showing similar results. Ns, p>0.05 (for comparison between WT and CD9 KO cells using an unpaired two-tailed Student´s t-test)”. This new information has been added to the corresponding figure legends”.

4) In Figs 3 and 6, no information is provided on replicates. This suggests only a single experiment was performed.

Response: We have clarified in the figure 3 legend that “Two experiments were performed, showing similar results (for the Western blots)”, and for the viral titers that “Three experiments were performed showing similar results. *p<0.05, **p<0.01, ***p<0.001, ****p<0.0001 (for comparison between cells transfected with the NT siRNA and the cells transfected with the NRP1 siRNA, using an unpaired two-tailed Student´s t-test)”.

For figure 6, the experiments in SW480 cells were performed twice, whereas the experiments in A549 cells were performed only once, because we run out of reagents. In any case, please note that, as indicated in the figure legend: ”for all images, the quantification of the average PLA signal per cell is shown to the right. For each condition, 10 images were acquired at 40× magnification, with each image containing between 80 and 120 cells”, therefore, around 1000 cells were taken into consideration for each condition.

5) In Fig 4, data is from 3 separate experiments (n = 3 biological replicates). However, data is only shown from one experiment. This is acceptable for the Western blots but not for the densitometry data. The data from all three experiments should be combined and analysed.

Reply: We concur with this reviewer´s comment. Therefore, we have combined the results from the three experiments in the quantification data, and we have indicated in the figure legend that “For quantifications, the mean and standard deviations, from the three Western blots performed are represented. *p<0.05, **p<0.01, ***p<0.001, ****p<0.0001 (for comparison between WT and CD9 KO cells using an unpaired two-tailed Student´s t-test)”.

6) In Fig 2, no statistical analysis is performed in panel A and yet the authors claim that there are differences in protein expression. In contrast, in panel B, mRNA data is analysed, found to be nonsignificant and stated to be not different. All data should be analysed in the same way. A similar qualitative description is also shown in Fig 6.

Response: For Figure 2A, we have specified that: “For the Western blots, two experiments were performed showing similar results. Representative blots from one out of two experiments are shown. For quantifications, the mean and standard deviations, of the three Western blots performed are represented. Ns, p>0.05, *p<0.05, **p<0.01, ***p<0.001, ****p<0.0001 (for comparison between WT and CD9 KO cells using an unpaired two-tailed Student´s t-test)”. According to this new statistical analysis, whereas we indicated that the expression of ACE2 was slightly decreased in the CD9 KO cells, these differences are not statistically significant, therefore, we have removed this conclusion from the text.

7) How was the antibody concentration chosen for the blocking experiment shown in Fig 1? Is it a saturating concentration, 50% saturation etc? In addition, are both the anti-CD9 and control antibodies used azide-free?

Reply: the antibody concentration was chosen based on a previous report in which the authors use the same commercial anti-CD9 antibody, and they show that blocking the cells with this antibody concentration reduces the infectivity of the coronavirus MHV, of an influenza virus, and of VSV particles pseudotyped with the VSV G protein, and with the spike proteins from the coronaviruses HCoV-229E, MERS-CoV, and SARS-CoV (Earnest, j. Virol. 2015). This new information has been introduced in the materials and methods and in the results sections, lines 292-297 in the current version.

We appreciate the reviewer’s comment regarding the potential for sodium azide to introduce a confounding factor in infection-blocking assays. However, after reviewing the technical specifications of both antibodies, we confirmed that each contains 0.09% sodium azide. Therefore, the observed decrease in infection in CD9-blocked cells can be attributed specifically to CD9 blockade rather than to azide-related effects. The azide content information can be found on the respective product webpages:

  • Anti-CD9 antibody: https://www.fishersci.es/shop/products/anti-cd9-clone-m-l13-bd/15848108
  • Isotype control antibody: https://www.thermofisher.com/antibody/product/Mouse-IgG1-kappa-clone-P3-6-2-8-1-Isotype-Control/14-4714-82

8) Previous work on tetraspanins and SARS-CoV-2 should be cited completely. There are some missing publications (Malla and Kamal, 2022 DOI: 10.2174/1381612828666220907105543 
Healy 2022 https://doi.org/10.1016/j.bbrc.2022.01.038 
Figueroa 2022 https://ecommons.luc.edu/luc_theses/4448 ).

Reply: According to this referee´s comment. We have introduced the two first references in the text, in lines 74 and 608, corresponding to numbers 22 and 49 in the revised version. The third reference is a master´s thesis, and although we respect the reviewer´s comment, we prefer not to include thesis documents among the listed references.

Reviewer 2 Report

Comments and Suggestions for Authors

The manuscript entitled „The Tetraspanin CD9 Facilitates SARS-CoV-2 Infection and Brings Together Different Host Proteins Involved inSARS-CoV-2 Attachment and Entry into Host Cells“ by Rivero et al. describes the role of CD9 in SARS-CoV-2 entry and its co-localization with proteins involved in SARS-CoV-2 virus attachment and entry. The findings are interesting, the results are conclusive.

There are a few remarks:

Materials and Methods:

1) Cells and viruses: Culture conditions for all cell lines are missing

2) Cells and viruses: Please describe the SARS-CoV-2 strain used in these experiments, what variant was used and is there sequencing data available?

3) Plasmids: The authors describe pCDNA3.1 as expression plasmid, and then pCAGGs plasmid is mentioned in the same paragraph? If CD9 was subcloned into pCAGGs plasmid, this should be described more clearly otherwise this paragraph is confusing.

4) SARS-CoV-2 infections: Infection could be described more detailed; how many cells were seeded and when were the cells infected? Which medium was used?

5) qRT-PCR: Please use reverse transcriptase instead of retrotranscriptase

6) qRT-PCR: Authors show primers for GAPDH but in the text they say that expression levels were normalized to actin expression levels? Please clarify. Minor observation: GAPDH primers are in antisense orientation, forward should be reverse and vice versa. There is one additional G in Rv primer sequence.

7) Immunofluorescence: Authors should mention how long slides were incubated with first and secondary antibodies (line 175 – 178, 180 - 181).

8) Duo Link: line 222 „amplified signals were washed…“- there should be a rewording since the slides and not the signals were washed

9) Results:

It is sometimes confusing that the term „CD9 KO cells“ was used for both, SW480 and A549 cells, maybe the cells could be renamed to SW480/CD9KO and A549/CD9KO or in a similar way.

10) Line 258: „…A549 cells which do not overexpress ACE2…“ should be changed to  „A549 cells which do not express ACE…“

11) Figure 1: Please explain why the neutralizing antibody against CD9 is effective after 24 hours, while the effect on SARS-CoV-2 virus titer in CD9 KO cells is delayed and seen only after 48 hrs. What could be the reason for this difference?

12) Figure 1: labelling „actina“ should be corrected to „actin“

13) Figure 1 caption: B) expression of CD9 was analyzed by flow cytometry – it is not mentioned which cells were used.

14) Figure 3 caption: line 335 should read: “…A549-ACE cells were silenced…”; line 339: A549 should be changed to A549-ACE cells

15) Line 312: should read „transcript levels“

16) Figure 5 and 6: images are of too low resolution

17) Figure 7B: there should be an arrow pointing towards CD9 to distinguish from the signal of the light chain of the CD9 antibody (this band could be indicated by an asterisk)

18) Supplementary: I can only find Supplementary Figure 1 as tiff-file but not the other figures 2-4, so the information is incomplete

Author Response

Dear Editors and reviewer,

Please find attached the revised version of the manuscript entitled "The Tetraspanin CD9 Facilitates SARS-CoV-2 Infection and Brings Together Different Host Proteins Involved in
SARS-CoV-2 Attachment and Entry into Host Cells " by Rivero et al., which we would like to resubmit to be further considered for publication at Viruses.

We would like to thank the referees and the editors for their helpful comments, and for giving us the opportunity to revise our paper. To address the issues raised by the reviewers, we have performed additional statistical analysis and made changes to the text where appropriate. Attached you will find a complete point-by-point reply to reviewers' comments.

We think that the manuscript has been significantly improved according to the reviewer´s comments. Therefore, we hope that in the light of the changes made, you will now find our manuscript suitable for publication at Viruses.

Looking forward to hearing from you soon, sincerely,

Marta L. DeDiego

REVIEWER 2

The manuscript entitled “The Tetraspanin CD9 Facilitates SARS-CoV-2 Infection and Brings Together Different Host Proteins Involved inSARS-CoV-2 Attachment and Entry into Host Cells“ by Rivero et al. describes the role of CD9 in SARS-CoV-2 entry and its co-localization with proteins involved in SARS-CoV-2 virus attachment and entry. The findings are interesting, the results are conclusive.

 There are a few remarks:

Materials and Methods:

1) Cells and viruses: Culture conditions for all cell lines are missing

Response: All the cells were grown at 37°C in air enriched with 5% CO2 using Dulbecco’s modified Eagle’s medium (DMEM, Gibco) supplemented with 10% fetal bovine serum (Gibco), and 50 mg/ml gentamicin (Gibco).

2) Cells and viruses: Please describe the SARS-CoV-2 strain used in these experiments, what variant was used and is there sequencing data available?

Response: According to this reviewer´s comment, we have clarified in the text that: “We used a SARS-CoV-2 strain isolated from Vero E6 cells, originated from a nasal swab from a patient infected in Madrid, Spain, at the beginning of 2020, termed SARS-CoV-2-MAD6 strain. This strain is similar to the B.1 variants and it was kindly provided by Prof. Luis Enjuanes, at Centro Nacional de Biotecnología, CNB-CSIC, Spain. Particularly, the SARS-CoV-2-MAD6 genome was sequenced and the sequence was identical to the SARS-CoV-2 Wuhan-Hu-1 isolate (SARS-CoV-2-WH1) genome (GenBank MN908947), with the exception of the silent mutation C3037>T, and two missense mutations: C14408>T (affecting nsp12) and A23403>G (leading to D614G in S protein), as previously described (Wang et al. Front Cell Infect Microbiol 2023)”

3) Plasmids: The authors describe pcDNA3.1 as expression plasmid, and then pCAGGs plasmid is mentioned in the same paragraph? If CD9 was subcloned into pCAGGs plasmid, this should be described more clearly otherwise this paragraph is confusing.

Response: we are sorry for the confusion. Indeed, the plasmid generated was the plasmid pcDNA3.1. Accordingly, we have modified the text in the revised manuscript as follows: “A polymerase II expression pcDNA3.1 plasmid encoding CD9 (GenBank accession number NM_022873) was generated by synthesizing the open reading frame of CD9 (IDT DNA, Inc) flanked by the unique restriction sites for BamHI and NheI and cloning this fragment in the pcDNA3.1 plasmid cut with the same enzymes”

4) SARS-CoV-2 infections: Infection could be described more detailed; how many cells were seeded and when were the cells infected? Which medium was used?

Response: according to this reviewer´s comment, we have clarified in the revised version that “WT and CD9 KO SW480 and A549-ACE2 cells were seeded in 24-well plates at 2x105 cells per well. The following day, cell monolayers (at 95% confluency) were infected with SARS-CoV-2 at a multiplicity of infection (MOI) of 0.2, in medium DMEM, (Gibco) supplemented with 2% fetal bovine serum (Gibco), and 50 mg/ml gentamicin (Gibco)”.

5) qRT-PCR: Please use reverse transcriptase instead of retrotranscriptase

Response: According to this reviewer´s comment, we have changed the word retrotranscriptase by reverse transcriptase

6) qRT-PCR: Authors show primers for GAPDH but in the text they say that expression levels were normalized to actin expression levels? Please clarify. Minor observation: GAPDH primers are in antisense orientation, forward should be reverse and vice versa. There is one additional G in Rv primer sequence.

Response: we apologize for the mistake. In fact, the primers we used to normalize the RNA levels are from actin mRNA. We have changed the sequence of the actin primers accordingly.

7) Immunofluorescence: Authors should mention how long slides were incubated with first and secondary antibodies (line 175 – 178, 180 - 181).

Response: Following this reviewer´s comment, we have specified in the revised version that the primary antibodies were incubated at 4ºC overnight, and the secondary antibodies were incubated at room temperature during 1 hour.

8) Duo Link: line 222 „amplified signals were washed…“- there should be a rewording since the slides and not the signals were washed

Response: according to this reviewer´s comment, we have changed the words “Amplified signals were washed” by the “slides were washed…”

9) Results: It is sometimes confusing that the term „CD9 KO cells“ was used for both, SW480 and A549 cells, maybe the cells could be renamed to SW480/CD9KO and A549/CD9KO or in a similar way.

Reply: according to this reviewer´s comment, we have named the cells SW480/WT, SW480/CD9 KO, A549-ACE2/WT or A549-ACE2/CD9 KO through the text.

10) Line 258: „…A549 cells which do not overexpress ACE2…“ should be changed to  „A549 cells which do not express ACE…“

Response: according to the reviewer´s suggestion, we have changed the word overexpress by express

11) Figure 1: Please explain why the neutralizing antibody against CD9 is effective after 24 hours, while the effect on SARS-CoV-2 virus titer in CD9 KO cells is delayed and seen only after 48 hrs. What could be the reason for this difference?

Response: We apologize for the mistake but, the viral titers shown in the anti-CD9 antibody-treated cells correspond indeed to 48 h post-infection. We cannot detect differences in viral titers at 24 h post-infection, between the cells treated with the control antibody and the CD9 antibody, as observed between the WT and CD9 KO cells. We have changed the time post-infection in the revised version accordingly.

12) Figure 1: labelling „actina“ should be corrected to „actin“

Reply: According to this reviewer´s point, this misspelling has been corrected in the revised version.

13) Figure 1 caption: B) expression of CD9 was analyzed by flow cytometry – it is not mentioned which cells were used.

Reply: the histograms in the submitted version (Figure 1B) corresponded to A549 cells. We have now clarified the cell line in the revised Figure 1B caption, and we have added histograms corresponding to SW480 wild-type and CD9 KO cells.

14) Figure 3 caption: line 335 should read: “…A549-ACE cells were silenced…”; line 339: A549 should be changed to A549-ACE cells

Reply: These minor edits have been corrected in the revised version

15) Line 312: should read „transcript levels“

Reply: following this reviewer´s comment, the word transcriptional levels has been changed by transcript levels in the revised version

16) Figure 5 and 6: images are of too low resolution

Reply:  We concur with this reviewer´s comment. Therefore, the resolution of figures 5 and 6 has been improved

17) Figure 7B: there should be an arrow pointing towards CD9 to distinguish from the signal of the light chain of the CD9 antibody (this band could be indicated by an asterisk)

Reply: according to this reviewer´s suggestion, we have introduced in the revised Figure 7B and * an an arrow to show the bands corresponding to antibody light chain and the CD9 protein, and we have clarified the arrow and * meanings in the figure legend

18) Supplementary: I can only find Supplementary Figure 1 as tiff-file but not the other figures 2-4, so the information is incomplete

Reply: We don´t know the reason why, but we can find the four supplementary Figures in the manuscript-supplementary file, which has been loaded in the webpage again.

Apart from the reviewer’s comments, we have added a materials and methods paragraph related to flow cytometry that we noticed it was missing (lines 184-192).

Round 2

Reviewer 1 Report

Comments and Suggestions for Authors

3) In Figs 1 and 2, experimental data is from 'triplicate wells'. This would appear to be a technical replicate, where an experiment is performed only once (a single biological replicate). This is inadequate and experiments must be performed in multiple biological replicates (n = 3 is typical).

Response: Following this referee´s comment, we have specified in Figures 1D and 1E, that: “(D and E) Error bars represent standard deviations (SD) of results of measurements performed in triplicate wells. Three experiments were performed showing similar results. 

4) In Figs 3 and 6, no information is provided on replicates. This suggests only a single experiment was performed.

Response: We have clarified in the figure 3 legend that “Two experiments were performed, showing similar results (for the Western blots)”, and for the viral titers that “Three experiments were performed showing similar results. *p<0.05, **p<0.01, ***p<0.001, ****p<0.0001 (for comparison between cells transfected with the NT siRNA and the cells transfected with the NRP1 siRNA, using an unpaired two-tailed Student´s t-test)”.

In both cases, the data from all three experiments should be suitably combined and statistical analysis performed. It is not acceptable to hide data in this way, by claiming 'similar experiments were performed'.

Author Response

1) Point 3) In Figs 1 and 2, experimental data is from 'triplicate wells'. This would appear to be a technical replicate, where an experiment is performed only once (a single biological replicate). This is inadequate and experiments must be performed in multiple biological replicates (n = 3 is typical).

Previous Response: Following this referee´s comment, we have specified in Figures 1D and 1E, that: “(D and E) Error bars represent standard deviations (SD) of results of measurements performed in triplicate wells. Three experiments were performed showing similar results. 

Point 4) In Figs 3 and 6, no information is provided on replicates. This suggests only a single experiment was performed.

Previous Response: We have clarified in the figure 3 legend that “Two experiments were performed, showing similar results (for the Western blots)”, and for the viral titers that “Three experiments were performed showing similar results. *p<0.05, **p<0.01, ***p<0.001, ****p<0.0001 (for comparison between cells transfected with the NT siRNA and the cells transfected with the NRP1 siRNA, using an unpaired two-tailed Student´s t-test)”.

In both cases, the data from all three experiments should be suitably combined and statistical analysis performed. It is not acceptable to hide data in this way, by claiming 'similar experiments were performed'.

Last Response: We appreciate the reviewer’s comment and understand the importance of providing information on biological and technical experimental replicates. Accordingly, we have performed new statistics analysis taking into account the data from the three experiments in figures 1D, 1E, 2B and 3B. Accordingly, we have changed the graphics in these figures and we have clarified in the corresponding figure legends that: “Error bars represent the means and standard deviations (SD) of results from three independent experiments including triplicate wells in each experiment”.

For figure 6, the experiments in SW480 cells were performed twice, whereas the experiments in A549 cells were performed only once, due to limited availability of reagents and the high cost of the technique. However, we did conduct a second independent experiment in SW480 cells, which yielded qualitatively similar results: positive signal for ICAM-1, NRP-1, and FURIN, and negative for ACE2 and TMPRSS2. Despite these consistent qualitative findings, the number of dots per cell varied between experiments and when we tried to pool the data, a high dispersion appeared. Thus, we believe it would be misleading to pool the quantitative data. In this context, the critical outcome of the Duolink assay is the presence or absence of red puncta indicating protein-protein proximity, rather than exact quantification. For the A549 cell line, the experiment was performed once, but we analyzed a large number of cells, as for SW480, analyzing approximately 1,000 cells across 10 representative images (ranging from 80 to 120 cells per image). The robust signal observed in A549 cells provides important validation in a second cell line, which we believe adds more value than repeating the experiment in SW480 cells alone for a third time. We hope this explanation clarifies our experimental approach and addresses the reviewer’s concern.

Reviewer 2 Report

Comments and Suggestions for Authors

The authors have sufficiently improved their manuscript which is, after a few minor corrections, now acceptable for publication.

  • Line 297:  authors missed to correct 24 hpi to 48 hpi
  • Please also correct retrotranscriptase to reverse transcriptase in Supplement Fig.1 and there is an inconsistent font type in the figure caption
  • Supplementary figure 4: please also change actina to actin

Author Response

Dear Editors and reviewer,

Please find attached the second revised version of our manuscript entitled "The Tetraspanin CD9 Facilitates SARS-CoV-2 Infection and Brings Together Different Host Proteins Involved in
SARS-CoV-2 Attachment and Entry into Host Cells " by Rivero et al., which we would like to resubmit to be further considered for publication at Viruses.

 We would like to thank the referees for their helpful comments, and for giving us the opportunity to revise our paper. To address the issues raised by the reviewers, we have performed additional statistical analysis and made changes to the text and figures where appropriate. Attached you will find a complete point-by-point reply to reviewers' comments.

We hope that in the light of the changes made, you will now find our manuscript suitable for publication at Viruses.

Looking forward to hearing from you soon, sincerely,

Marta L. DeDiego

REVIEWER 2:

The authors have sufficiently improved their manuscript which is, after a few minor corrections, now acceptable for publication. There are a few remarks:

1) Line 297:  authors missed to correct 24 hpi to 48 hpi

Response:  We apologize for this mistake that has been corrected in the revised version (line 297)

2) Please also correct retrotranscriptase to reverse transcriptase in Supplement Fig.1 and there is an inconsistent font type in the figure caption

Response: According to this referee´s comment, we have changed the word retrotranscriptase to reverse transcriptase, and the fond type in Supplementary Figure 1 legend.

3) Supplementary figure 4: please also change actina to actin

Response: According to this reviewer´s comment, we have changed the word actina for actin in Supplementary Figure 4.